# Seven Days of Bismuth-Based Quadruple Therapy Is as Effective for the First-Line Treatment of Clarithromycin-Resistant Confirmed *Helicobacter pylori* Infection as 14 Days of Bismuth-Based Quadruple Therapy

**DOI:** 10.3390/jcm11154440

**Published:** 2022-07-30

**Authors:** Sang-Gon Moon, Chul-Hyun Lim, Hee-Jun Kang, Arum Choi, Sukil Kim, Jung-Hwan Oh

**Affiliations:** 1Division of Gastroenterology, Department of Internal Medicine, Eunpyeong St. Mary’s Hospital, The Catholic University of Korea, Seoul 03312, Korea; sgmoon83@gmail.com (S.-G.M.); heejun87@gmail.com (H.-J.K.); ojh@catholic.ac.kr (J.-H.O.); 2Department of Preventive Medicine, College of Medicine, The Catholic University of Korea, Seoul 06591, Korea; dyemelody@gmail.com (A.C.); sikimmd@catholic.ac.kr (S.K.)

**Keywords:** *Helicobacter pylori*, clarithromycin, bismuth, duration of therapy, point mutation

## Abstract

Background/Aims: Point mutations in the 23S ribosomal RNA gene have been associated with *Helicobacter pylori* (*H. pylori*) clarithromycin resistance and bismuth-based quadruple therapy (BQT) is one of the options for the treatment of clarithromycin-resistant *H. pylori*. Current *H. pylori* treatment guidelines recommend BQT for 10–14 days. This study aims to compare the eradication extents according to 7-day and 14-day BQT treatment for treatment-naïve clarithromycin-resistant confirmed *H. pylori* infection. Methods: We retrospectively investigated treatment-naïve *H. pylori* infection cases from March 2019 to December 2020, where patients were treated with BQT. Clarithromycin resistance was identified with a dual-priming oligonucleotide-based multiplex polymerase chain reaction method. We reviewed a total of 126 cases. Fifty-three subjects were treated with a 7-day BQT regimen (7-day group), and 73 subjects were treated with a 14-day BQT regimen (14-day group). We evaluated the total eradication extent of the BQT and compared the eradication extents of the two study groups. Results: Total eradication extent of *H. pylori* was 83.3% (105/126). The eradication extents of the two groups were as follows: 7-day group (81.1% (43/53)), 14-day group (84.9% (62/73), *p* = 0.572) by intention-to-treat analysis; 7-day group (95.6% (43/45)), 14-day group (92.5% (62/67), *p* = 0.518) by per-protocol analysis. The moderate or severe adverse event extents during the eradication were 30.2% (16/53) in the 7-day group and 19.2% (14/73) in the 14-day group (*p* = 0.152). Conclusions: The 7-day BQT regimen was as effective as the 14-day BQT regimen in the eradication of treatment-naïve clarithromycin-resistant *H. pylori* infection.

## 1. Introduction

The *Helicobacter pylori* (*H. pylori*) infection has been reported to be associated with diverse gastrointestinal diseases including chronic gastritis, peptic ulcer disease, gastric mucosa-associated lymphoid tissue lymphoma and gastric adenocarcinoma, and several extra-gastric diseases, including iron deficiency anemia, primary immune thrombocytopenia and vitamin B12 deficiency anemia [1,2]. The eradication of *H. pylori* has been reported to be beneficial for treatment or prevention of HP- associated diseases [3,4,5,6]. The importance of eradication has been emphasized, and indications of eradication have been expanded. The American College of Gastroenterology (ACG) clinical guidelines recommend treating all *H. pylori* infection cases [3]. Japanese guidelines for managing *H. pylori* infection also indicate that the entire *H. pylori*-infected patient represents the treatment target [7]. Nevertheless, the eradication rate of *H. pylori* has decreased worldwide [8], at least in part due to the increasing antibiotic resistance of *H. pylori*, and antibiotic resistance is regarded as one of the most important causes of treatment failure [9,10,11]. Among several antibiotics used to treat *H. pylori*, clarithromycin resistance is considered a major factor the most important for treatment failure, and resistance is steadily rising. Clarithromycin resistance in the American, European, and Asian regions is reported to be 8–18%, 25–31%, and 5–27%, respectively, and Korean clarithromycin resistance is reported to be 17.8–37.0% at present [12,13,14,15].

Dual-priming oligonucleotide-based multiplex polymerase chain reaction (DPO-PCR) has a sensitivity and specificity sufficient enough to be compared with the sensitivity and specificity of the culturing method for detecting clarithromycin resistance. The DPO-PCR method detects point mutations in the 23S RNA gene of *H. pylori*, which causes resistance to certain antibiotics, including clarithromycin. Several mutations have been discovered (A2142C, A2142G, A2143G, A2144T, T2717C, and C2694A), and clinical studies reported high *H. pylori* eradication rates following identification of clarithromycin resistance with the DPO-PCR method [16,17,18].

Maastricht V/Florence guidelines suggest bismuth-based quadruple therapy (BQT), consisting of a proton pump inhibitor (PPI), bismuth subcitrate, metronidazole, and tetracycline, is a therapeutic option for clarithromycin-resistant *H. pylori* infection. According to the guidelines, BQT is one of the first-line choices of treatment where local clarithromycin and metronidazole resistance extents are over 15%, which is the case in Korea [8].

The 2013 Korean *H. pylori* treatment guidelines recommend prescribing BQT for 7–14 days, and many physicians still use 7-day BQT as the first-line therapy for clarithromycin-resistant *H. pylori* [19]. Current clinical guidelines, including Maastricht V/Florence guidelines, ACG guidelines, and Korean *H. pylori* treatment guidelines, recommend 10–14 days administration of BQT [3,8,12]. There has been no study convincing enough to provide definitive evidence for the appropriate duration of BQT. In other words, there has been no consensus on the treatment duration of BQT, especially when it is used as first-line therapy for clarithromycin-resistant confirmed *Helicobacter pylori* infection. Since BQT is quite unpleasant to take for some patients, the appropriate administration duration of BQT is of practical importance.

We designed a study that used the DPO-PCR method to assess the eradication extent of BQT as first-line therapy in *H. pylori* infection that was identified as resistant to clarithromycin. We compared the regimen’s eradication extents and adverse events according to different treatment durations.

## 2. Materials and Methods

### 2.1. Study Subjects

We retrospectively reviewed the medical records of 807 *H. pylori* infection cases in Eunpyeong St. Mary’s Hospital, Catholic University, Seoul, the Republic of Korea, from March 2019 to December 2020, in which BQT was used. We only included treatment-naïve cases, which were also proven to be resistant to clarithromycin using the DPO-PCR method. We only included A2143G and A2142G point mutations because they are the dominant mutations in Korea, and we were not able to detect the other mutations in our hospital [11]. We assessed the point mutations (A2142G and A2143G) for clarithromycin resistance by using U-TOPTM HPy-ClaR Detection Kit (SEASUN BIOMETRIALS, Daejeon, Korea).

We excluded following cases from our study; (1) patients who have undergone any form of gastric surgery; (2) pregnant or breast-feeding women; (3) patients who were prescribed probiotics or other drugs to reduce the side effects of the BQT regimen during the eradication; (4) patients who were under age 18.

The BQT regimen used in this study consisted of a double dose of PPI; metronidazole, 500 mg three times a day; bismuth subcitrate, 300 mg four times a day; and tetracycline, 500 mg four times a day. Several different PPIs (lansoprazole, 30 mg; pantoprazole, 40 mg; or rabeprazole, 20 mg; illaprazole, 10 mg) were permitted.

*H. pylori* eradication was assessed with a ^13^C-urea breath test (UBT; UBiTkit; Otsuka Pharmaceutical Co., Ltd., Tokyo, Japan) at least 4 weeks after completing the eradication regimen [8]. We prescribed no PPIs or histamine receptor blockers after completing the BQT regimen until the study patients underwent UBT to minimize any possible effects on the UBT results. Follow-up loss cases were defined when patients did not complete the UBT at least 12 weeks after completing the regimen. We still included patients who completed the UBT 12 weeks after completing the BQT regimen. We did not fix the limit of UBT timing as long as the UBT was performed at least 4 weeks after completing the BQT regimen.

We included patients in a per-protocol analysis (PP analysis) who successfully underwent the whole BQT regimen and were examined with a UBT. We included patients who could not complete the BQT regimen (poor compliance) or follow-up loss cases in an intention-to-treat (ITT) analysis. Poor compliance was considered when the study subjects did not take more than 85% of the medication.

We divided the study subjects into two groups according to treatment duration, the 7-day group (subjects prescribed 7 days of BQT) and the 14-day group (subjects prescribed 14 days of BQT). There were 53 subjects in the 7-day group and 73 subjects in the 14-day group. We depended solely on history taking when we identified the study subjects’ treatment experience of *H. pylori*. We identified a total of 112 subjects’ UBT results.

### 2.2. Outcome Measures and Compliance

We conducted a comparison of the eradication extent of each treatment group. We also analyzed the eradication extents according to mutations and compared adverse events of each study group. We checked the compliance and adverse events when the study subjects’ eradication was examined with the UBT.

We clinically analyzed significant adverse events of the study subjects. We classified adverse events as mild, moderate, or severe, according to the degree of interruption to ordinary daily living. When study subjects experienced certain kinds of adverse events that did not cause any significant impact on daily living, we classified them as mild adverse events. When study subjects’ daily living was interrupted, we classified adverse events as moderate. When study subjects had to stop the BQT regimen because of the events or when hospitalization was required to manage adverse events, we classified the adverse events as severe. We only included moderate or severe adverse events in the statistical analysis because mild adverse events had no clinical impact.

### 2.3. Statistical Analysis

*H. pylori* eradication extent was analyzed using ITT and PP analyses. Categorical variables were analyzed using a Chi-square test or Fisher’s exact test. Continuous variables were analyzed using independent *t*-test. A Chi-square test with Yate’s correction for continuity was applied where a statistic value was 0% or 100%. Multivariate logistic regression was performed using treatment results as the dependent variable and age, sex, alcohol intake, smoking status, hypertension, diabetes mellitus, treatment duration, and mutation types as covariates. Results are presented as on odds ratio with a 95% confidence interval. Statistical significance was defined as a *p*-value less than 0.05. All statistical analyses were calculated using SPSS software version 23.0 for Windows (IBM Corp., Armonk, NY, USA).

### 2.4. Ethics Statement

This study was approved by the Institutional Review Board of Eunpyeong St. Mary’s Hospital, Catholic University, Seoul (IRB number: PC20RISI0227). The requirements for written informed consent were waived because anonymous data were collected. This study followed the ethical principles of the Declaration of Helsinki.

## 3. Results

### 3.1. Characteristics of the Study Population and Total Eradication Extent

Baseline characteristics of all study subjects are shown in Table 1. The average age of the two study groups was different. The average age of the 7-day group was 62.2 years, and the average age of the 14-day group was 55.8 years. According to the eradication indications suggested by Korean *H. pylori* treatment guidelines, functional dyspepsia, except admissive indications, was the most indicated cause for eradication, followed by peptic ulcer diseases (Table 2) [12].

There were 105 subjects with an A2143G mutation (83.3%), 18 subjects with an A2142G mutation (14.3%), and 3 subjects who had both A2143G and A2142G mutations (2.4%). We identified 112 subjects’ UBT results among 126 study subjects (88.9%). The total eradication extent was 83.3% (105/126) by ITT analysis and 93.8% (102/112) by PP analysis.

### 3.2. Eradication Extents According to the Treatment Duration of BQT

Among the 53 subjects in the 7-day group, we identified 45 subjects’ UBT results. The eradication extent of the 7-day group was 81.1% (43/53) by ITT analysis and 95.6% (43/45) by PP analysis. Among the 73 subjects in the 14-day group, we identified 67 subjects’ UBT results. The eradication extent of the 14-day group was 84.9% (62/73) by ITT analysis and 92.5% (62/67) by PP analysis. The eradication difference was statistically insignificant by both ITT analysis and PP analyses (Figure 1).

### 3.3. Eradication Extents According to Mutations

We analyzed eradication extents according to mutations in the two study groups. The eradication extents of the A2142G mutation in the 7-day group were 66.7% (4/6) by ITT analysis and 80.0% (4/5) by PP analysis. The eradication extents of the A2143G mutation in the 7-day group were 84.8% (39/46) by ITT analysis and 100% (39/39) by PP analysis. The eradication extent with both mutations in the 7-day group was 0% (0/1) by ITT and PP analyses. There was no statistically significant difference in eradication extents among different mutation profiles in the 7-day group (Table 3).

The eradication extent of the A2142G mutation in the 14-day group was 91.7% (11/12) by ITT analysis and 100% (11/11) by PP analysis. The eradication extent of the A2143G mutation in the 14-day group was 84.7% (50/59) by ITT analysis and 90.9% (50/55) by PP analysis. The eradication extent of the subjects with both mutations in the 14-day group was 50% (1/2) by ITT and 100% (1/1) by PP analysis. There was no statistically significant difference in eradication extents among different mutation profiles in the 14-day group (Table 3). In addition, we found no clinical factor deemed associated with eradication rates in multivariate analysis (Table 4).

### 3.4. Adverse Events According to the Therapeutic Duration

We analyzed moderate and severe adverse events in the two study groups. Of the total study subjects, 23.8% (30/126) were identified as having experienced a moderate or severe adverse event during eradication. The moderate or severe adverse event extents were 30.2% in the 7-day group and 19.2% in the 14-day group (*p* = 0.152). The most frequently reported adverse event was nausea, followed by vomiting. Although there were differences in a few specific details, there was no significant difference between the two study groups regarding adverse events on the whole (Table 5).

## 4. Discussion

We found that the 7-day group was not different from the 14-day group in the eradication of treatment-naïve clarithromycin-resistant *H. pylori* infection cases. Moreover, both study groups’ moderate-to-severe adverse event extents were not significantly different.

Our study is consistent with other studies that found no difference in eradication e tents and adverse events extents when BQT was administered for different durations. 

Recently, Shin et al. reported no significant differences in *H. pylori* eradication extent between the 7-day administration and 10- to 14-day administration of the BQT regimen, which was used as second-line treatment [20]. Kim et al. reported that twice-daily BQT for 7 days was no worse than the conventional 14-day BQT regimen in terms of efficacy and safety, although the BQT was used as second-line therapy and there was no antibiotic susceptibility test such as the DPO-PCR test in their retrospective study [21]. In their Cochrane meta-analysis, Yuan et al. reported no significant difference in relative risks for *H. pylori* persistence according to treatment duration [22]. Kim et al., in their recent review article on *H. pylori* eradication, noted that whether the eradication rate improves when the treatment period is extended from 7 to 14 days is unclear [9].

We identified a high adverse events extent with the BQT regimen; 23.8% (30/126) in the subjects who reported they had experienced a moderate or severe adverse event. We also identified no difference in adverse events of BQT between the two study groups. The extent of moderate or severe adverse events of BQT was 30.2% in the 7-day group and 19.2% in the 14-day group. (*p* = 0.152) The treatment duration seemed unrelated to the occurrence or severity of clinically significant adverse events. In a study comparing 10-day BQT and 14-day BQT in Italy, Dore et al. reported there was no meaningful difference in adverse events of BQT between 10-day and 14-day treatment groups, and they also reported adverse events occurred in the early days of eradication. According to the study, treatment duration was not associated with the occurrence of adverse events [23]. In a prospective study comparing the efficacy of BQT as second-line therapy between 7-day and 14-day administration in Korea, severe side effects were reported in each group, 15.3% and 21.8% (*p* = 0.243), respectively [24]. 

The Maastricht V/Florence consensus report recommends that physicians should use BQT for 14 days unless a 10-day BQT regimen is proven locally effective [8]. The ACG treatment guidelines for *H. pylori* suggest that BQT be administered for 10–14 days when used as the first-line treatment [3]. Although Korean guidelines for *H. pylori* treatment in 2013 allowed for a 7-day administration of BQT, the 2020 Korean guidelines changed the recommendation of BQT duration from 7–14 days to 10–14 days [12,25]. The Toronto consensus statements for *H. pylori* recommend physicians should use BQT for 14 days regardless of whether physicians use BQT as the first-line or second-line therapy [5]. Those guidelines, however, have a shallow quality of evidence as to duration and lack robust prospective studies as evidence for treatment duration. In other words, there have been no convincing controlled trials to decide proper durations of BQT, especially when physicians use BQT as the first-line therapy. Clearly, well-designed prospective studies that compare 7-day and 14-day BQT administration are necessary.

In deciding on the treatment duration of the BQT regimen, whether the *H. pylori* strain has metronidazole resistance might be more critical than the presence of clarithromycin resistance because metronidazole resistance may be overcome with a longer treatment duration [8,26]. However, we cannot currently acquire information on the metronidazole resistance test. The availability of the metronidazole test in the clinic might help in this context.

We identified a high eradication extent of BQT when used in a tailored manner based on the DPO-PCR method. Tailored therapy using DPO-PCR is generally believed to be superior to standard treatment [26,27]. The ideal way to secure high eradication of *H. pylori* is a culture-based sensitivity test before the start of treatment. Unfortunately, it is rarely available in the clinic. Our study confirmed that tailored therapy based on the DPO-PCR test may lead to a high eradication extent.

We also identified that the A2143G mutation is the dominant mutation of clarithromycin resistance. The incidence of the A2143G mutation was 83.3%, and that for the A2142G mutation was 14.3% in our study, which coincided with the known distribution of clarithromycin-resistance mutations [16,25]. We also identified that the A2143G mutation is the dominant mutation of clarithromycin resistance. The incidence of the A2143G mutation was 83.3%, which coincided with the known distribution of clarithromycin-resistance mutations in Korea (71.4%) [17]. A2143G mutation is known to be one of the most common mutations in other countries as well (Italy, USA, etc.) [16,25,28]. However, we found that a specific mutation type was not related to eradication rate, when using BQT (Table 3). We presume that specific clarithromycin-resistance profiles might not affect the treatment results because the BQT regimen does not include clarithromycin. There was also no other clinical factor that is clearly related with eradication rate in multivariate analysis (Table 4). 

Our study had some limitations. This study was retrospectively undertaken at a single center and the two study groups were not identical in baseline characteristics. This study was retrospectively undertaken at a single center, so our study lacks randomization. Especially, the two study groups were not identical with respect to age. The average age of the 7-day group was higher than that of the 14-day group. It seemed that many patients over the age of 65 years in our hospital wanted to take 7, rather than 14, days of BQT, considering they were already on many other medications. Further randomized study is necessary in this context.

Our study has the following strengths. The BQT regimen was used for treatment-naïve *H. pylori* infection cases, and we only included *H. pylori* infection cases in which definite resistance to clarithromycin was confirmed with the DPO-PCR method. To our knowledge, there are no published studies evaluating the eradication extent of BQT as the first-line therapy according to treatment duration in patients confirmed to be infected by resistant strains of *H. pylori.*

## 5. Conclusions

The 7-day BQT regimen was as effective as the 14-day BQT regimen for eradication of treatment-naïve clarithromycin-resistant confirmed *H. pylori* infection. Physicians could consider a 7-day BQT regimen as the first-line option for the treatment of clarithromycin-resistant confirmed *H. pylori* infection.

## Figures and Tables

**Figure 1 jcm-11-04440-f001:**
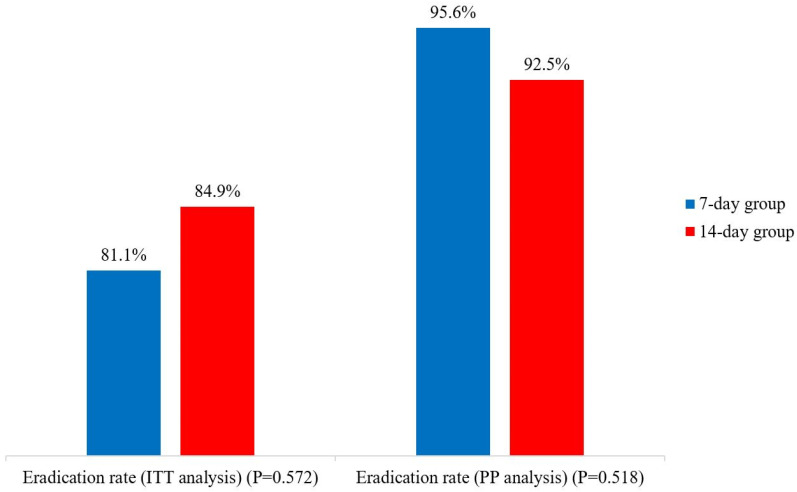
Eradication extents of the two study groups. ITT analysis (Intention-to-treat analysis), PP analysis (Per-protocol analysis).

**Table 1 jcm-11-04440-t001:** Basic characteristics.

	7-Day Group	14-Day Group	*p*-Value
Number	53	73	
Age (year, mean)	62.2	55.8	0.009
Sex (%)			
Male	17 (32.1%)	24 (32.9%)	0.925
Female	36 (67.9%)	49 (67.1%)	0.925
Diabetes Mellitus (%)	7 (13.2%)	4 (5.5%)	0.129
Hypertension (%)	15 (28.3%)	15 (20.5%)	0.313
Body mass index (mean)	24.8	24.4	0.581
Cigarette smoking (%)	11 (20.8%)	14 (19.2%)	0.827
Alcohol intake (%)	18 (34.0%)	34 (46.6%)	0.156

**Table 2 jcm-11-04440-t002:** Comparisons of the two study groups by eradication indications *.

Eradication Indications	7-Day Group	14-Day Group	*p*-Value
Functional dyspepsia	20 (37.8%)	52 (71.2%)	<0.001
Peptic ulcer diseases	17 (32.1%)	22 (30.1%)	0.816
Family history of gastric cancer	1 (1.9%)	8 (11.0%)	0.051
After resection of gastric adenoma	2 (3.8%)	1 (1.4%)	0.382
After resection of early gastric cancer	0 (0.0%)	1 (1.4%)	1.000
Long-term low dose aspirin user with a history of peptic ulcer diseases	0 (0.0%)	1 (1.4%)	1.000
Atrophic gastritis **	52 (98.1%)	64 (87.7%)	0.032
Intestinal metaplasia **	9 (17.0%)	9 (12.3%)	0.461

* Eradication indications suggested by Korean College of *Helicobacter* and Upper Gastrointestinal research in 2020; ** admissive indications [12].

**Table 3 jcm-11-04440-t003:** Eradication rates of the two study groups.

		A2142G	A2143G	Both	*p*-Value *
7-day group	ITT	4/6 (66.7%)	39/46 (84.8%)	0/1 (0.0%)	0.270
	PP	4/5 (80.0%)	39/39 (100.0%)	0/1 (0.0%)	0.218
14-day group	ITT	11/12 (91.7%)	50/59 (84.7%)	1/2 (50.0%)	0.530
	PP	11/11 (100.0%)	50/55 (90.9%)	1/1 (100.0%)	0.677

ITT (Intention-to-treat analysis); PP (Per-protocol analysis); * comparison between A2142G mutation group and A2143G mutation group.

**Table 4 jcm-11-04440-t004:** Clinical factors associated with *H. pylori* eradication.

Variable	Multivariate Analysis		
	OR *	95% CI **	*p*-Value
Age	1.019	0.970–1.071	0.450
Sex			
Men	Reference		
Women	3.327	0.768–14.424	0.108
Alcohol intake			
Non-drinker	Reference		
Drinker	1.050	0.338–3.263	0.933
Smoking state			
Non-smoker	Reference		
Smoker	3.091	0.647–14.776	0.157
Hypertension			
No	Reference		
Yes	1.978	0.598–6.546	0.264
Diabetes mellitus			
No	Reference		
Yes	0.173	0.013–2.293	0.183
Treatment duration			
7-day	Reference		
14-day	0.811	0.280–2.347	0.699
Type of mutation			
A2142G	Reference		
A2143G	0.879	0.199–3.883	0.865

* OR, odds ratio; ** CI, confidence interval.

**Table 5 jcm-11-04440-t005:** The adverse event profile of the study subjects.

	7-Day Group	14-Day Group	*p*-Value
Nausea	12 (22.6%)	9 (12.3%)	0.125
Vomiting	5 (9.4%)	6 (8.2%)	0.812
Sense of weakness or tiredness	2 (3.8%)	7 (9.6%)	0211
Dizziness	2 (3.8%)	5 (6.8%)	0.457
Somnolence	1 (1.9%)	6 (8.2%)	0.126
Dyspepsia and bloating sense	3 (5.7%)	7 (9.6%)	0.421
Diarrhea	0 (0%)	4 (5.5%)	0.224
Dysgeusia	1 (1.9%)	3 (4.1%)	0.482
Abdominal pain or discomfort	0 (0%)	5 (6.8%)	0.138
Xerostomia	0 (0%)	5 (6.8%)	0.138
Poor oral intake	0 (0%)	4 (5.5%)	0.224
Headache	1 (1.9%)	3 (4.1%)	0.482
Epigastric discomfort or heartburn	4 (7.5%)	6 (8.2%)	0.908
Itchiness	0 (0%)	2 (2.7%)	0.622
Skin rash	0 (0%)	2 (2.7%)	0.622
Insomnia All adverse events	1 (1.9%) 16 (30.2%)	0 (0%) 14 (19.2%)	0.872 0.152

## Data Availability

The datasets generated and/or analyzed during the current study are not publicly available due to our IRB policy but are available from the corresponding author upon reasonable request.

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
