# Peer review of "Seven Days of Bismuth-Based Quadruple Therapy Is as Effective for the First-Line Treatment of Clarithromycin-Resistant Confirmed Helicobacter pylori Infection as 14 Days of Bismuth-Based Quadruple Therapy"

_jcm, 2022, doi:10.3390/jcm11154440_

Round 1

Reviewer 1 Report

Sang-Gon Moon's manuscript describes an interesting cohort evaluating the benefit of using BQT for 7 or 14 days. 

The major problem is that the study is designed as a superiority study and not an equivalence/non-inferiority study and some information is missing. 

Introduction/Discussion: discuss the benefit of using only TSA-guided treatment.

Prefer passive voice throughout the manuscript.

Methods: did the author's hospital look for other mutations?

Methods: justify the exclusion criterion of "3 doses".

Methods/results: authors should discuss mild adverse events as they could be the reason for discontinuation for most treated patients.

Methods: how did the authors consider correction of multiple tests?

The titles of paragraphs should appropriately represent their content, especially when the results show a non-significant association.

Results: The authors indicated that efficacy could be associated with the presence/absence of certain mutations. Indeed, they did not demonstrate this and should compare 7 vs 14 after stratification by mutation (not the other way around).

REsutls: it is very surprising that fewer adverse events occur in the longer treatment group. Justify.

italicize "et al."

Table 4: what are "all" adverse events?

Author Response

The major problem is that the study is designed as a superiority study and not an equivalence/non-inferiority study and some information is missing. 

  • Thank you for your kind pointing out. Our study is a retrospective study where we reviewed the treatment results of 7-day and 14-day BQT regimen on treatment-naïve H. pylori patients of our center. We did not intend to check out superiority, equivalence, or inferiority directly in our study. We are working on a prospective study where we compare 7-day and 14-day BQT regimen on H. pylori infected patients now. We are looking forward to checking out superiority, equivalence, or inferiority in our prospective study.

Introduction/Discussion: discuss the benefit of using only TSA-guided treatment. (discuss the benefit of using only antibiotic susceptibility testing-guided treatment.)

  • Thank you for your kind pointing out again. We mentioned the benefit of DPO-PCR based treatment of H. pylori on Page 2 lines 50~53. Recently published several studies have shown better treatment results of DPO-PCR based eradication regimen than empirical treatment.

“Several mutations have been discovered (A2142C, A2142G, A2143G, A2144T, T2717C, and C2694A), and clinical studies have reported high H. pylori eradication rates following identification of clarithromycin resistance with the DPO-PCR method.”

  • We commented about this in discussion section on Page 8 line 264~269 in the revised manuscript.

“We identified a high eradication extent of BQT when used in a tailored manner based on the DPO-PCR method. Tailored therapy using DPO-PCR is generally believed to be su-perior to standard treatment.22,23 The ideal way to secure high eradication of H. pylori is a culture-based sensitivity test before the start of treatment. Unfortunately, it is rarely avail-able in the clinic. Our study confirmed that tailored therapy based on the DPO-PCR test may lead to a high eradication extent.

Prefer passive voice throughout the manuscript.

  • Thank you for your kind pointing out again. We changed a few sentences where passive voice sentences seemed unclear.

Methods: did the author's hospital look for other mutations?

  • Thank you for your kind pointing out again. We did not look for other clarithromycin mutations except A2142G and A2143G. We only included A2143G and A2142G point mutations because they are the dominant mutations in Korea, and we were not able to detect the other mutations with the detection kit which is in use in our hospital. We included the additional comments in the revised manuscript on Page 2 line 83~87.

“We only included A2143G and A2142G point mutations because they are the dominant mutations in Korea, and we were not able to detect the other mutations in our hospital. 11 We assessed the point mutations (A2142G and A2143G) for clarithromycin resistance by using U-TOPTM HPy-ClaR Detection Kit (SEASUN BIOMETRIALS, Daejeon, Korea).”

 Methods: justify the exclusion criterion of "3 doses".

  • Thank you for your kind pointing out again. We reviewed our data and refined exclusion criteria and PP analysis inclusion criteria, and we re-wrote 2.1. ‘study subject’ on page 3 line 105-108.

“We included patients in a per-protocol analysis (PP analysis) who successfully un-derwent the whole BQT regimen and were examined with a UBT. We included patients who could not complete the BQT regimen (poor compliance) or follow-up loss cases in an intention-to-treat (ITT) analysis. Poor compliance was considered when the study subjects did not take more than 85% of the medication.”

Methods/results: authors should discuss mild adverse events as they could be the reason for discontinuation for most treated patients.

  • Thank you for your kind pointing out again. We did not include mild adverse events because we only thought of clinically significant adverse events as meaniful. As it were mentioned in the manuscript, if a patient had to stop because of an adverse events, which would be considered as moderate or severe. Having to stop the medicine due to adverse events means that the adverse events were of moderate or severe degree. There were no patients with mild adverse events who stopped the regimen.

Methods: how did the authors consider correction of multiple tests?( the Authors have performed multiple tests with a unique set of data. They have to consider the inflation of the alpha risk (pvalue) by using appropriate statistical methods.)

  • Thank you for your kind pointing out again. We ran two tests in our study;(1) DPO-PCR which checked specific mutations causing clarithromycin resistance. it was used for screening test to determine the study patients.; (2) UBT which checked treatment success or failure after study patients’ completing the study regimen. If there had been three tests in our study, that might have led to inflation of the alpha risk as you mentioned above. However, we ran two tests, and we concluded those tests were not associated with inflation of the alpha risk. We consulted on this with a statistics expert of our center.
  •  

The titles of paragraphs should appropriately represent their content, especially when the results show a non-significant association.

  • Thank you for your kind opinion. We changed the title as “7 days of Bismuth-based Quadruple Therapy is as effective for the first-line treatment of Clarithromycin-resistant confirmed Helicobacter pylori infection as 14 days of Bismuth-based Quadruple Therapy.”

Results: The authors indicated that efficacy could be associated with the presence/absence of certain mutations. Indeed, they did not demonstrate this and should compare 7 vs 14 after stratification by mutation (not the other way around).

  • Thank you for your kind pointing out again. Table 3 (Eradication rates of each of the two mutation groups) shows comparisons of eradication rates according to the mutations. We also applied continuity correction in our statistics as advised by second reviewer from your journal. In that comparison, we showed mutation types were not clearly associated with eradication rate. We also found no statistically significant results in multivariate analysis (table4). We added the result of multivariate analysis in the revised manuscript. That’s why we concluded a specific mutation did not affect eradication rates. We guessed that clarithromycin mutations do not affect eradication rates because BQT regimen did not include clarithromycin.
  •  

REsutls: it is very surprising that fewer adverse events occur in the longer treatment group. Justify.

  • Thank you for your kind pointing out again. As you commented, we found fewer adverse events in the longer treatment group, but it was not statistically significant. Regarding adverse events, we guess that it depends on probability. In other words, main factor of whether adverse events take place or not is matter of probability, not of treatment duration. So, we discussed on it on Page 7 line 235~243.

“The treatment duration seemed unrelated to the occurrence or severity of clinically significant adverse events. In a study comparing 10-day BQT and 14-day BQT in Italy, Dore ET AL. reported there was no meaningful difference in adverse events of BQT between 10-day and 14-day treatment groups, and they also reported adverse events occurred in the early days of eradication. According to the study, treatment duration was not associated with the occurrence of adverse events. In a prospective study comparing the efficacy of BQT as second-line therapy between 7-day and 14-day administration in Korea, severe side effects were reported in each group, 15.3% and 21.8% (P = 0.243), respectively.”

italicize "et al."

  • Thank you for your kind pointing out again. We italicized them all.

Table 4: what are "all" adverse events?

  • Thank you for your kind pointing out again, but we did not use the word of “all” adverse events. In case the “all” that you meant was the “all” in table 4, we changed “Any” adverse events to “All” adverse events.

Reviewer 2 Report

In the present retrospective study, performed on 126 patients with clarithromycin-resistant H. pylori, determined by molecular biology methods, Moon et al showed that a 7-days bismuth containing quadruple therapy (BQT) is equivalent to 14-days BQT. Main comments:

1) Please add p values in Table 2

2) In Table 4, since there are several zero values, a Chi square test with Yates correction is more proper.

3) Page 7 lines 243-244: this statement should be supported by literature (see De Francesco V et al, Ann Intern Med 2006).

4) Lack of randomization should be acknowledged as a further limitation.

Author Response

1) Please add p values in Table 2

- Thank you for your kind pointing out. we changed the table and added newly calculated p-values in table 2 as you mentioned.

2) In Table 4, since there are several zero values, a Chi square test with Yates correction is more proper.

       - Thank you for your kind pointing out again. We reviewed the statistics of our study, and Yates correction was applied in the revised manuscript.

3) Page 7 lines 243-244: this statement should be supported by literature (see De Francesco V et al, Ann Intern Med 2006).

-  Thank you for your kind pointing out again. We checked the study of De Francesco, and we identified that the distribution patterns of clarithromycin resistance mutations vary among different countries. In Korea, A2143G mutation is known to be the most frequently identified mutation. It differs from Italy and other countries as well. We changed the above-mentioned sentences and quoted a study on Korean distribution of clarithromycin mutations on Page 8 line 273 ~ 276 of the revised manuscript.

4) Lack of randomization should be acknowledged as a further limitation.

- Thank you for your kind pointing out again. We also acknowledged our study lacked randomization. It was inevitable because our study was retrospectively designed. We changed and added a few sentences on Page 8 line 285~286 of the revised manuscript.

“This study was retrospectively undertaken at a single center, so our study lacks randomization….. Further randomized study is necessary in this context.”

Round 2

Reviewer 1 Report

The manuscript has been improved according to my previous comments.

Author Response

We deeply appreciated for your kind and delicate review and comments of the article.

Reviewer 2 Report

Regarding point 2, Chi square with Yates correction should be mentioned in Methods paragraph, not in table 2.

Regarding point 3, A2143G is still one of the most common mutation even in Italy.

Author Response

Regarding point 2, Chi square with Yates correction should be mentioned in Methods paragraph, not in table 2.

Thank you again for your comment. We added a new sentence ‘Chi-square test with Yate’s correction for continuity was applied where a statistic value was 0% or 100%’ in the statistical analysis section of the second revised manuscript, and we deleted the related mention which had been located below table2 and table3.

Regarding point 3, A2143G is still one of the most common mutation even in Italy.

Thank you again for your kind comment. We added the sentence as below according to your comments with reference (Ann Intern Med 2006;144:94-100, Curr Opin Infect Dis 2017;30:489-497)

"A2143G mutation is known to be one of the most common mutations in other countries as well (Italy, USA, etc.)"